# Fundamental Limits on Earth-like Exoplanet Imaging with Large Telescopes Employing Laser Tomographic Adaptive Optics Systems: A Comparative Analysis of LGS AO and LTAO Systems

**Keran Deng [1], Jian Huang [2,*] and Ke Wang [1]**

1   College of Physics and Electronic Engineering, Chongqing Normal University, Chongqing 401331, China; dken2140@163.com (K.D.); maons@live.cn (K.W.)
2   Chongqing Key Laboratory of Green Design and Manufacturing of Intelligent Equipment, School of Mechanical Engineering, Chongqing Technology and Business University, Chongqing 400067, China
*   Correspondence: huangjian_honor@163.com

**Abstract:** Exoplanet imaging with high-contrast imaging adaptive optics systems, though challenging, is a promising path toward the characterization of terrestrial planets. We analyzed the fundamental limitations associated with the direct imaging of terrestrial exoplanets around low-mass stars with Extremely Large Telescopes using laser tomographic adaptive optics (LTAO) and derived the post-coronagraph image shape in the focal plane from LTAO systems. Additionally, the fundamental limitation of direct imaging was found to come from unseen spatial frequencies during tomographic reconstruction. Through the provision of optimization strategies for laser guide star (LGS) asterisms, based on the post-coronagraph image contrast, we aimed to assist in the design of LTAO systems for Extremely Large Telescopes, resulting in a six-fold improvement in the LTAO post-coronagraph image plane at 0.1 arcseconds.

**Keywords:** adaptive optics; high-contrast imaging; laser guide stars; laser tomographic adaptive optics

## 1. Introduction

The imaging of potentially habitable exoplanets with high-contrast imaging adaptive optics (HCIAO) systems is a promising, but challenging, method through which to characterize terrestrial planets around stars using next-generation Extremely Large Telescopes [1]. However, the flux contrast between a rocky planet and its solar-type star is $10^{-10}$ and lies beyond the fundamental limits of high-contrast imaging with ground-based Extremely Large Telescopes and their coronagraphic extreme adaptive optics (AO) systems [2]. The most accessible rocky planets are near low-mass M-type dwarf stars, which present the following factors: firstly, the contrast separation between the planet and the star is nearly $10^{-8}$; secondly, the angle between the planet and the star (considering the habitable zone) is within the reach of future Extremely Large Telescopes in the near-IR range [3]. Meanwhile, planets around low-mass M-type stars present unavoidable challenges to direct imaging, due to these stars' brightness. The extreme AO system is based on the traditional AO system, which accurately controls the residual error and aims to directly image exoplanets. Compared with traditional AO systems, the density of deformable mirror (DM) actuators and the wavefront sensor (WFS) detection frequencies of extreme AO systems are greatly increased, presenting higher requirements for the brightness of the guide star (e.g., apparent magnitude < 9 mag) [3]. The potentially habitable rocky planets around M-type stars cannot be imaged with extreme AO systems because their parent M-type dwarf stars are not bright enough to ensure that each WFS sub-aperture has a sufficiently high signal-to-noise

ratio [1]. The number of such low-mass M-type stars (e.g., apparent magnitude > 12 mag) exceeds 50% in the solar neighborhood within 5 pc (16.3 light years), which is beyond the wavefront detection capability of extreme AO [4].

The W. M. Keck Observatory's next-generation adaptive optics (NGAO) facility first proposed direct imaging of exoplanets around faint stars via guiding with its laser guide stars (LGS) [5]. In addition, NGAO would be used alongside four LGS beacons to perform tomography of the 3D mapping atmosphere in a narrow volume around the science field. Such a system is commonly described as using laser tomographic adaptive optics (LTAO) [6]. In practice, the typical Strehl Ratio (SR) that one can obtain using LTAO is in the range of 50% or less in the optical and near-IR wavelengths for the 25 m telescope GMT [7], which is too far away to reach the contrast level of 10 billion required for imaging an Earth-like planet around a Sun-like star, with an apodized mask special coronagraph [8,9]. In this article, we mainly discuss one of the fundamental limits of LTAO systems with large telescopes for high-contrast imaging.

The primary purpose of LTAO is to reduce focal anisoplanatism (FA). FA error is caused by the finite altitude of the artificial laser beacon, which is the single largest wavefront error term for the Large Telescope LGS AO system. The residual global FA error variance corrected with LTAO, known as the tomographic error, depends on the LGS's asterism [10]. The tomographic error variance is usually used as the evaluation standard for the optimal arrangement of LGSs in the classic AO system. However, the geometry of the LGS asterism also sets fundamental limitations for the tomography of high-contrast imaging of an on-axis target for extreme AO, namely, an unseen spatial frequency [11]. Unseen spatial frequency errors cause speckle noise at the corresponding image plane position. For a given turbulence profile, the unseen spatial frequency only depends on the guide stars' positions, as follows: the more distant the guide stars, the smaller the separation areas affected in the image plane, which eventually leads to changes in imaging contrast within a given area range of interest [12]. The optimizing strategy of the LGS asterism is focused on the imaging contrast. The purpose of HCIAO system optimization is to obtain the optimal contrast of a specific position on the image plane, that is, where the planet appears [13,14]. A lot of research, quantified using SR or residual variance, describes the ultimate limits of LTAO systems [15–17]. Unlike classical AO systems, residual variance or SR can only obtain image plane energy peak information, but not image plane contrast profile information [18]. With straightforward scaling from the phase spatial power spectrum density (PSD), we derived the post-coronagraph image shape in the focal plane. This technique was first introduced by Rigaut [19], who considered the AO system as a linear one, and analyzed the error PSD in the spatial frequency domain. Guyon, upon finding the correspondence between speckle amplitude and the above PSD, derived the post-coronagraph contrast with natural guide star cases [20], while Neichel first presented the residual phase spatial PSD of the LTAO system [11]. We followed Neichel's approach, with additional developments in the resulting scaling laws to optimize the LGS asterism, based on the contrast of the focal plane and the fundamental prediction limits of high-contrast imaging on Earth-like exoplanet imaging with Large Telescopes using an LTAO system [21]. The remainder of this paper is organized as follows: An overview of assumptions is presented in Section 2. In Section 3, the optimal method is validated using a numerical simulation; the total error, including the tomographic error, is calculated to test the validity of the proposed method. Section 4 quantifies AO system performance and post-AO imaging contrast when using laser tomographic technology. In Section 5, the single LGS AO and LTAO exoplanet detection capabilities are compared, and the unseen frequency in the LTAO system is discussed to optimize the contrast of certain imaging positions. In Section 6, conclusions and future study prospects are presented.

## 2. Assumptions

In order to generalize the research contents, we set the following assumptions:

- We consider that the sensing wavelength is the same as the imaging wavelength, ignoring differential chromatic effects.
- Independent tip/tilt correction was performed.
- We do not consider the impacts of telescope characteristics, such as the central obscuration ratio, primary mirror segmentation, or secondary mirror supports, regarding the high-contrast imaging capabilities [22].
- The coronagraph is assumed to be a "perfect coronagraph", which is used to suppress the diffraction pattern of stellar light [23].
- Only the AO halo remains after the stellar coronagraph, and it can hide planet in its speckle noise. The speckle noise and static aberrations are neglected. Aliasing error is also neglected, and can be removed with the pyramidal WFS of a spatially filtered Shack–Harmann sensor [24,25].

### 3. The Residual Phase Variance of Laser Tomographic Adaptive Optics and LGS Constellation Optimization

Laser tomography is used to simultaneously measure several wavefront slopes from LGSs at different angular positions in the sky and conduct the 3D mapping of a turbulent wavefront. Thus, a realistic simulation was performed for a 30 m telescope, $800 \times 800$ pixels at the pupil plane, with an imaging wavelength $\lambda$ of 2.2 μm. We considered a circular unobstructed telescope. The atmospheric phase screens were generated with an object-oriented adaptive optics toolbox (OOMAO) based on MATLAB (MathWoks Inc., Natick, MA, USA) [26]. We used a minimum mean square error (MMSE) reconstructor to reconstruct the wavefront of an LGS asterism, which was evenly located on a certain angular ring in the sky, and each wavefront $\varphi$ was derived from WFS slopes $s$, as follows [26]:

$$\varphi = C_{\varphi s} C_{ss}^{-1} s \tag{1}$$

where $C_{ss}$ is the slope covariance matrix and $C_{\varphi s}$ is the cross covariance between the wavefront and the slopes. Furthermore, we consider same number of Shack–Hartmann WFSs, corresponding to the number of LGSs, with 80 sub-apertures across the telescope diameter. The atmospheric parameters are listed in Table 1.

**Table 1.** Simulated parameters of atmosphere.

| Phase Screen | 1 | 2 | 3 |
|---|---|---|---|
| Height [m] | 300 | 5000 | 12,000 |
| Speed [m/s] | 10 | 5 | 2 |
| $C_n^2$ fraction | 0.5 | 0.3 | 0.2 |
| $r_0$ @ 500 nm [m] | | 0.10 | |
| Telescope diameter [m] | | 1–30 | |
| Guide star magnitude | | 8 | |

The MMSE wavefront estimate was obtained using Equation (1). The wavefront phase was removed from the NGS wavefront $\varphi_0$ to obtain the residual phase error. The wavefront residual error is related to the LGS constellation. We changed the number of LGS asterisms and sky angles to obtain the optimal LGS constellation, in the form of the residual error root mean square (RMS), as shown in Figure 1.

The 30 m telescope achieved minimal focal anisoplanatism RMS with five laser guide stars uniformly distributed on a circle with a radius of 35 arcseconds. This distribution closely resembles the laser guide star configuration of the NFIRAOS system from the 30 m telescope [27].

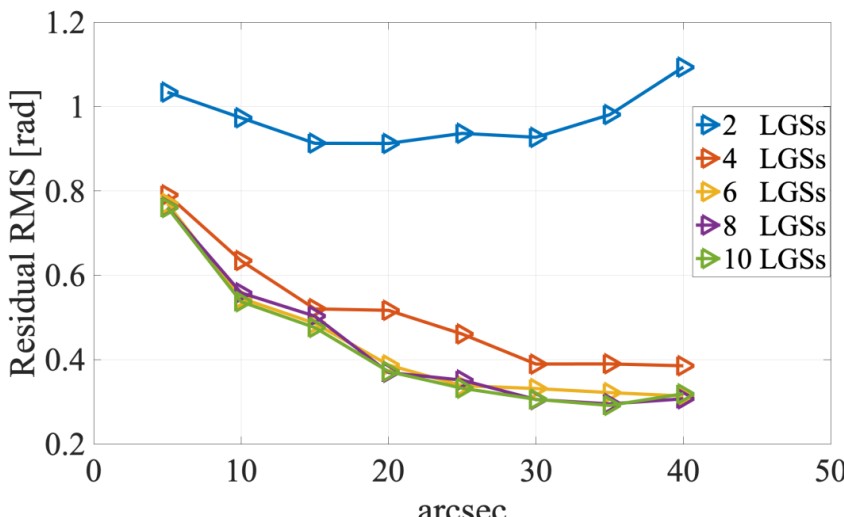

**Figure 1.** Residual phase error RMS via MMSE construction of LTAO.

## 4. Limiting Post-Coronagraphic Contrast under Minimum Mean Square Error Control of Laser Tomographic Adaptive Optics

With the previous developments, we obtained the minimal residual wavefront phase from the optimal LGS constellation. Using the Taylor expansion of the point spread function (PSF), including phase-only error, we assumed a perfect coronagraph that was used to suppress the diffraction pattern of stellar light, leaving a halo term that is essentially the power spectrum of the wavefront aberrations. Considering our assumptions from Section 2, the post-coronagraphic point spread function (PSF) relates to variance as follows [20]:

$$I_{pc}(\rho) = \sigma_k^2 [I(\rho - k\lambda) + I(\rho + k\lambda)] \tag{2}$$

where $I_{pc}(\rho)$ is the post-coronagraphic PSF in the focal plane, and $\rho$ is the coordinate of the image plane; a more complete treatment can be found in reference [28], which provides an analytic expression for long-exposure post-coronagraphic images through turbulence, but was not adopted here; $\sigma_k^2$ is the variance of the spatial model indexed via frequency $k$. Using the Parseval theorem, the post-coronagraphic PSF is proportional to post-AO residual PSD, and the post-coronagraphic contrast is defined as follows:

$$c(\rho) = \frac{I_{pc}(\rho)}{I(0)} \tag{3}$$

where $I(\rho)$ is the AO-corrected PSF.

Under the above theorem, we converted the minimal residual wavefront phase to PSD through the statistical average of the square modulus of the residual phase; finally, we obtained the contrast limit of the 30 m telescope due to the tomographic error, as shown in Figure 2.

In Figure 2, the various markers denote the following: red circles—selected known self-luminous directly imaged (DI) exoplanets with known H-band contrasts [1]; colored filled circles—the predicted reflected light flux ratios for a known star within 5 pc. Different colors correspond to different magnitudes, with one-to-one correspondence with the color coordinates on the right side [29]. Various curves denote the following: 8–10 m ground-based curve, GPI—5-sigma post-processed contrast curve for H-band integral field spectrograph mode, 1 h integration [30]; 30 m telescope goal curve—the possible range of near-IR post-processed detection limits for extreme adaptive optics in wavefront sensing capability guiding regarding the natural star. The AO PSD includes the servo lag and the photon noise on the wavefront sensor from Guyon's work [1]. Top dotted line: V = 12 for a cool star observed from the ground; middle dotted line: V = 8 for a median warm star;

bottom dotted line: V = 5 for a warm star. Results are for a 30 m telescope with an LTAO curve, for the possible range of near-IR post-coronagraphic detection limits with only the cone effect, 5-sigma SNR, and 2 h integration on a V = 12 host star.

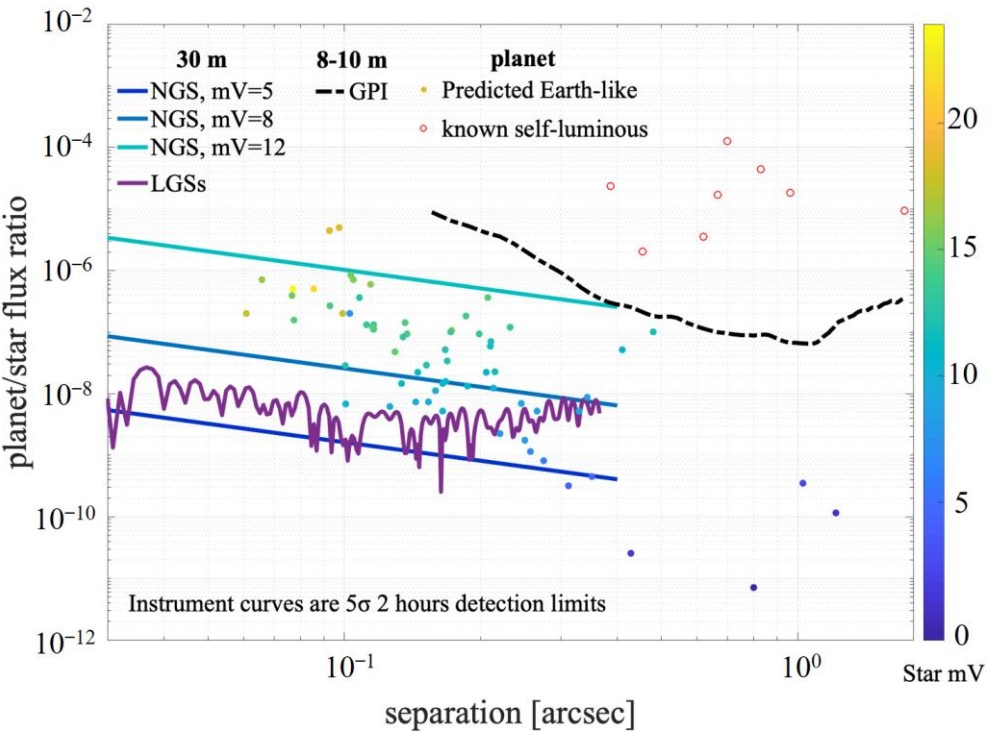

**Figure 2.** Light flux ratio between a planet with its parent star versus angular separation.

## 5. Discussion

### 5.1. Comparison of Single LGS AO and LTAO Exoplanet Detection Capabilities

The limitation of LGS AO for high-contrast imaging depends on the type of AO system and its geometry. One of the important issues is FA error, which depends on the altitude and the relative strength of the turbulence layer. The power spectrum of the FA PSD $W_{cone}(k)$ is calculated with respect to the atmospheric phase power spectrum, as follows:

$$W_{cone,h}(k) = 0.0229 \left(k^2 + L_0^{-2}\right)^{-\frac{11}{6}} r_0^{-\frac{5}{3}} \times \left[1 - 2\gamma A(k) + \gamma^2\right] \quad (4)$$

where the $A(k)$ is the Airy function. Based on the definition of PSF contrast, the PSF contrast after AO and the coronagraph of FA equates to the following:

$$C_{cone}(k) = W_{cone,h}(k) \cdot \frac{1}{D^2} \quad (5)$$

We compared the pupil residual phase due to the FA effect for a single-sodium LGS and an 8 LGS asterism, and the results are presented in Figure 3, wherein the left side shows the residual phase of the sodium LGS and phase RMS, and the right side shows the 8 LGS asterisms. The FA effect was significantly reduced with LTAO, and the residual FA RMS decreased from 1.547 rad to 0.540 rad.

Figure 4 compares the FA-PSF (after coronagraph) contrast of multiple LGS LTAOs and a single-sodium LGS AO with a 30 m telescope. Comparing the PSF contrast, as shown in Figure 4, it was found that the PSF contrast was reduced from $10^{-5}$ to $10^{-7}$ nearly 0.1″ after LTAO to reduce the focusing unequal halo effect.

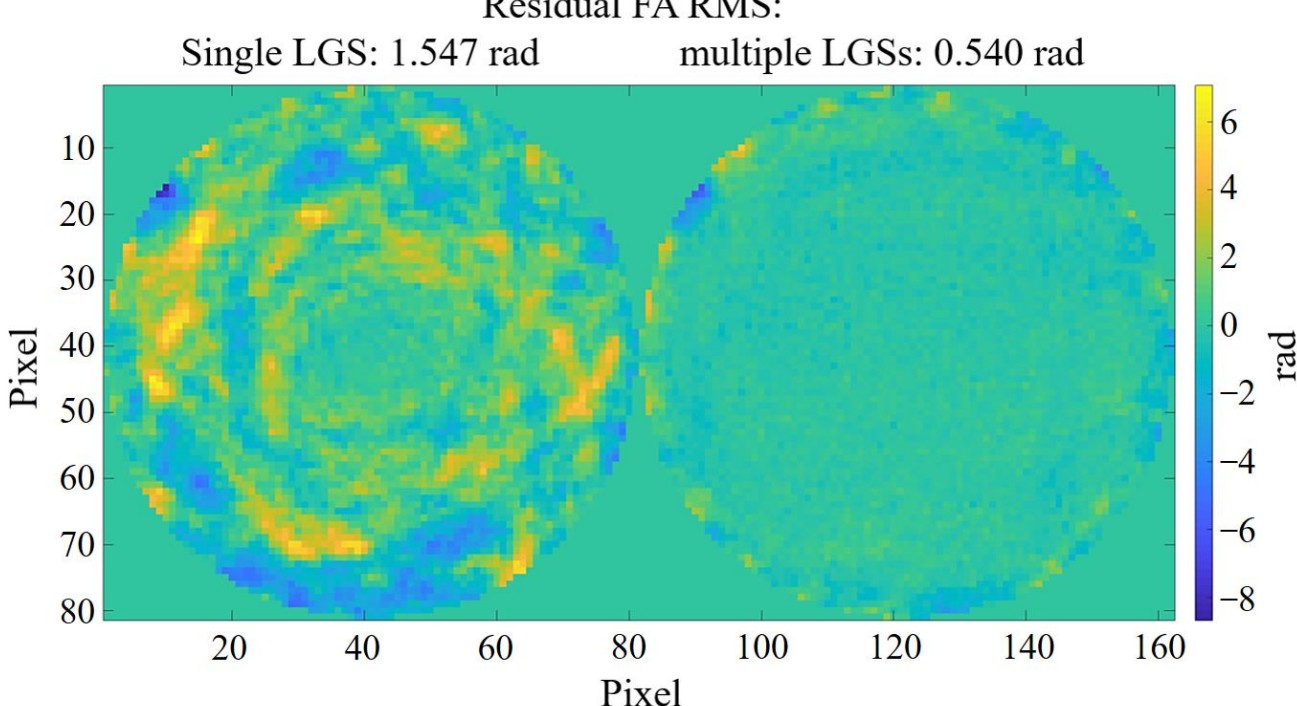

**Figure 3.** FA errors introduced with a single-sodium LGS AO and an LTAO. Left: single-sodium LGS AO; right: multiple LGS AOs.

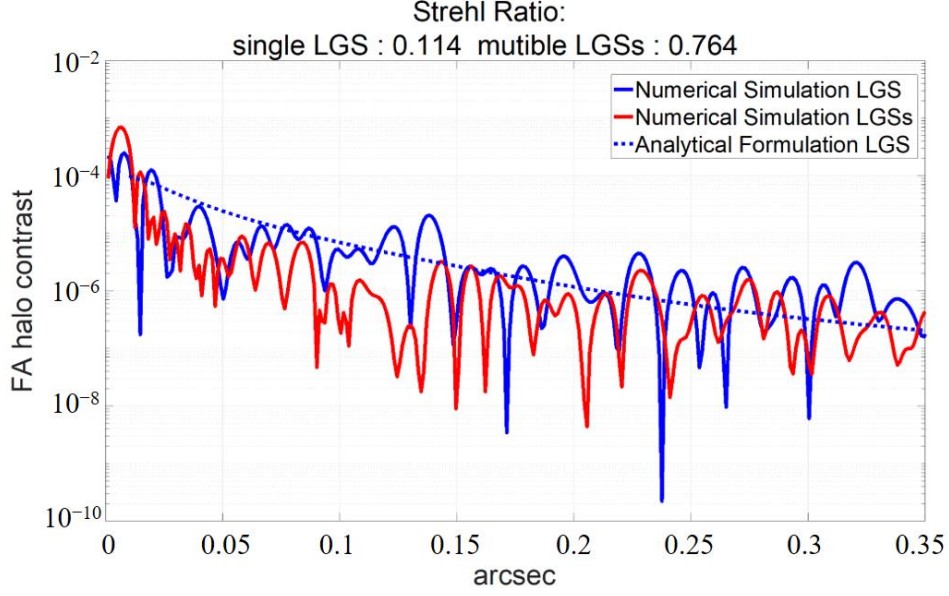

**Figure 4.** The relationship between the FA PSF contrast and the angular distance of the image plane. The solid blue line is the FA PSF contrast, introduced by the sodium beacon in the simulation experiment; the blue dotted line is the theoretically calculated single-sodium beacon FA PSF contrast; the solid red line is the residual FA PSF contrast of LTAO.

*5.2. Analysis of the LTAO Unseen Spatial Frequency Error*

The geometry of the LGS constellation sets the fundamental limitations of the unseen tomographic spatial frequency. As shown in Figure 5, an unseen spatial frequency leads to a decreasing trend of direct imaging capabilities in areas beyond 0.1 arcseconds in the image plane. We refer to these areas as "unseen areas", as they are not sensed with the WFSs. For a given turbulence profile, the unseen areas only depend on the guide stars'

positions, that is, the more distant the guide stars, the smaller the separation areas affected in the image plane, which eventually leads to changes in imaging contrast in a given area range of interest. We can re-optimize the LGS constellation based on the image plane and imaging contrast from the above characteristics, instead of through RMS.

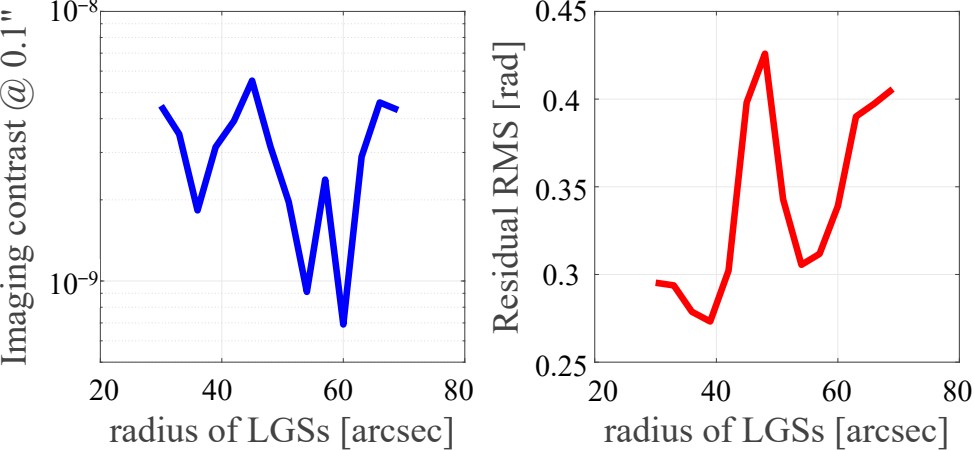

**Figure 5.** The relationship between imaging contrast and the angular radius of the LGS constellation around 0.1 arcseconds in the image plane (**left**); the globe residual variance is shown for comparison (**right**).

The relationship between imaging contrast (SNR = 5, 2 h detection duration) and the angular radius of an LGS constellation around 0.1 arcseconds in the image plane is shown in Figure 5. The globe residual variance is also shown for comparison. Specifically, in optimizing a single-ring asterism for future 30 m telescopes, residual variance is commonly used as a criterion. However, this criterion only provides a global description of energy concentration in the image plane and does not capture detailed intensity information at other positions of interest in the image plane. These specific positions, such as at 0.1 arcseconds of the image plane, may potentially be regions where exoplanets could appear. The 30 m telescope achieves the minimal imaging contrast, at 0.1 arcseconds in the imaging plane, under the LGS constellation with a 60-arcsecond radius. However, the residual variance is not the minimum, so the LGS constellation cannot be optimized based on the residual variance. The covariance matrices mentioned in Equation (1) are computed using the slopesLinearMMSE class in OOMAO code. The LGSs are propagated through the atmosphere and the telescope to the wavefront sensor. The LMMSE wavefront estimate is obtained by multiplying the slopesLinearMMSE object and the Shack–Hartmann object. The wavefront phase is removed from the NGS wavefront to obtain the zero-piston residual wavefront, as well as the residual wavefront error RMS.

Only considering the post-coronagraphic halo contrast, the 30 m telescope with a laser tomographic adaptive optics system (LGS optimal asterism) demonstrates the ability to achieve detection limits of $7 \times 10^{-10}$ of the flux of the host star (the V magnitude is 12) at separations of approximately 0.1 arcseconds in the focal plane. However, the imaging contrast through minimal residual RMS is only $3 \times 10^{-9}$. Figure 6 illustrates the atmospheric parameters adopted in this study for considering the diverse actual turbulence profiles. The weights and altitudes of the turbulence layers were derived from four nights of multi-aperture scintillation sensor (MASS) measurements atop Mauna Kea in 2002 [31].

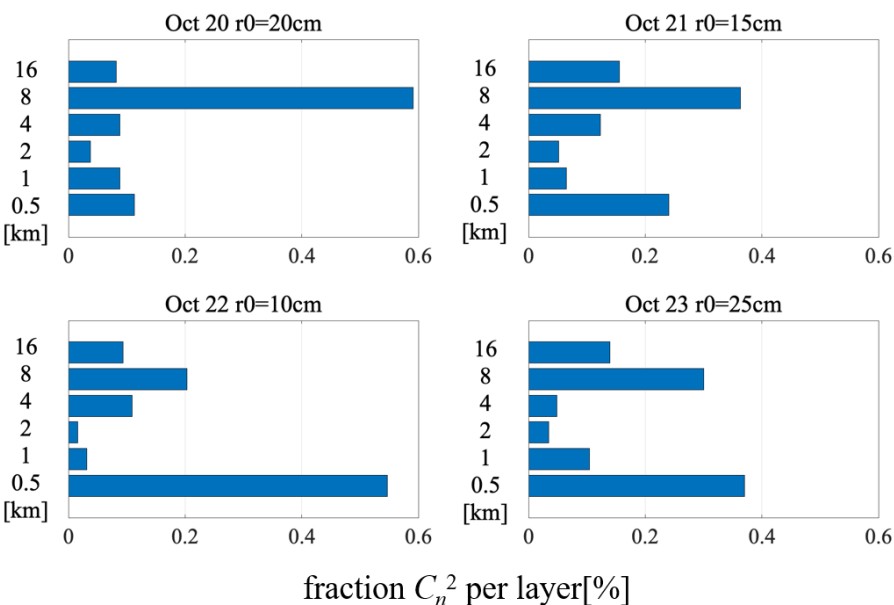

**Figure 6.** Turbulence profile derived from Mauna Kea observation.

The minimal imaging contrast associated with the optimal residual phase error RMS is depicted in Figure 7. This figure highlights how the optimal radius of an LGS for minimal contrast at specific positions in the image plane relates to the fractional $C_n^2$ distribution across different turbulence heights. We acknowledge the variability observed among the atmospheric profiles, including variations in turbulence distribution with altitude, and the caution that needs to be taken against extrapolating these results to other telescope sites. While our analysis provides valuable insights based on the available data, the diverse nature of atmospheric conditions underscores the need for further research to ensure the robustness and generalizability of our findings across different observational conditions.

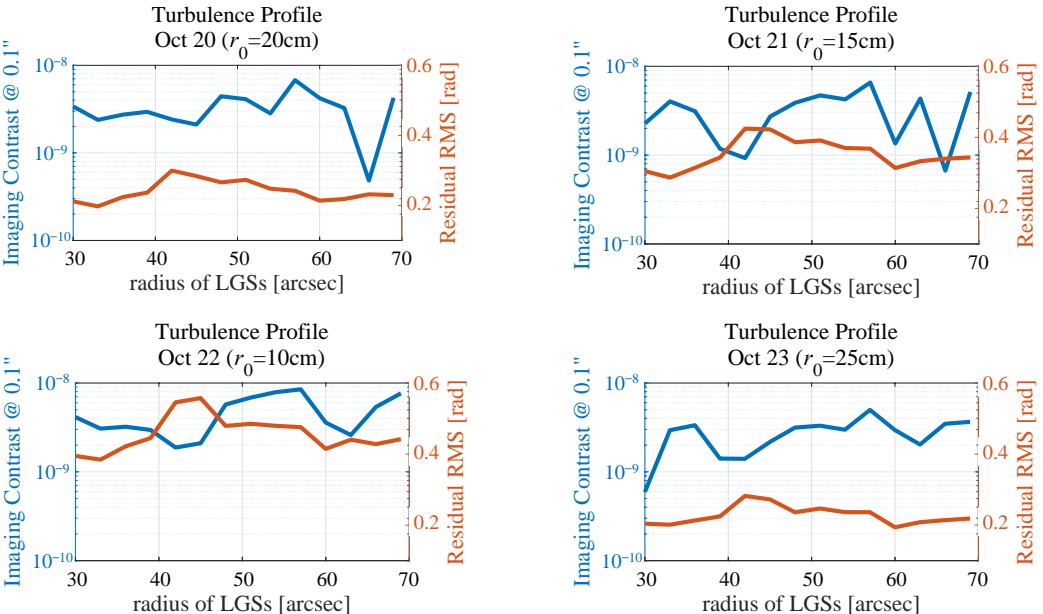

**Figure 7.** The imaging contrast at $0.1''$ in the image plane versus the angular radius of the LGS constellation (blue curve and blue coordinates); the whole wavefront error RMS is shown for comparison (red curve and red coordinates).

## 6. Conclusions

In this study, we investigated the limitations of ground-based telescopes in directly imaging terrestrial exoplanets around faint stars using LGS AO and LTAO systems. Our analysis focused on optimizing the HCIAO system to achieve optimal contrast at specific positions on the image plane, particularly at exoplanet locations. While our analysis primarily examined circular ring configurations for the LGS constellation, we acknowledge the potential benefits of exploring alternative asterism geometries, such as a uniform grid of LGS beacons. However, certain asterism configurations may be limited by the requirement of an unobstructed aperture. We recognize the impacts of central obstructions on real telescopes and acknowledge that our conclusions are based on a limited dataset from Mauna Kea. Thus, the generalization of our results to other potential 30 m telescope sites may vary. Overall, our analysis demonstrates significant potential for contrast improvement in LTAO-based HCIAO systems using 30 m telescopes, with an anticipated six-fold improvement in image plane contrast at 0.1 arcseconds. These findings contribute to the advancement of our understanding of high-contrast imaging techniques and pave the way for future enhancements in exoplanet detection capabilities.

**Author Contributions:** Conceptualization, K.D. and J.H.; methodology, K.D.; validation, J.H. and K.W.; formal analysis, K.W.; investigation, K.D.; resources, K.D. and K.W.; data curation, J.H. and K.W.; writing—original draft preparation, K.D.; writing—review and editing, J.H.; visualization, K.W.; supervision, K.W.; project administration, K.D.; funding acquisition, K.D., J.H. and K.W. All authors have read and agreed to the published version of the manuscript.

**Funding:** This study was funded by the National Natural Science Foundation of China (grant nos. 62305038 and 62105047); the Science and Technology Research Program of the Chongqing Municipal Education Commission (grant nos. KJQN202200567 and KJQN202100546); the Doctoral 'Through train' Research Project of Chongqing (grant no. CSTB2022BSXM-JSX0006); and the Chongqing Normal University Initiation Grant (grant no. 202006000161/02060404).

**Institutional Review Board Statement:** The study did not require ethical approval.

**Informed Consent Statement:** The study did not involve humans.

**Data Availability Statement:** Data is unavailable due to privacy.

**Conflicts of Interest:** We, the undersigned, declare that this manuscript is original, has not been published before, and is not currently being considered for publication elsewhere, with no interest in pursuing publication elsewhere. This statement of interest is declared on behalf of all authors.

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
