# Peer review of "Fundamental Limits on Earth-like Exoplanet Imaging with Large Telescopes Employing Laser Tomographic Adaptive Optics Systems: A Comparative Analysis of LGS AO and LTAO Systems"

_photonics, doi:10.3390/photonics11040338_

Round 1

Reviewer 1 Report

Comments and Suggestions for Authors

Review of "Fundamental Limits on Earth-Like Exoplanet Imaging with Large Telescopes by Laser Tomography Adaptive Optics System"

This is an interesting paper on characterizing the limiting contrast of exoplanet imaging while employing laser-tomographic adaptive optics on future 30-m class telescopes.  The outcome is a demonstration of good improvement in contrast with laser-tomography, and recommendations on laser asterism configurations to combat focal anisoplantism, which is certainly a worthwhile advancement to report on, and will be a welcome addition to the AO-design literature.  Current, public simulation software and well-defined analyses are employed to make predictions.  The descriptions are generally clear, properly referenced and make a good case that laser-tomographic techniques are a useful supplement to the toolbox of contrast improvement.  A minor weakness of the analysis seems to be that the diversity of atmospheric conditions input to the simulations is fairly limited; I read this as four different nights sampled in the early 2000's on Maunakea.  Also, some defence of the asterism chosen to study, and so comment on a limitation of the analysis should be made. But the authors do not overstate conclusions based on these choices, so it's not a serious issue - it would just benefit from some very brief further discussion, which I recommend the authors do.  Some suggestions to make this more clear are made in the detailed comments to follow.  All other comments are minor English-usage flaws or standardized nomenclature, which I point out below while reading through carefully, but this was not exhaustive.  I point them out merely to check during editing, and prior to careful English editing and publication in the journal.

Detailed comments:

Title: "Fundamental Limits on Earth-Like Exoplanet Imaging with Large Telescopes Employing Laser Tomographic Adaptive Optics Systems"  A suggestion is to replace the word "by" with "Employing"  to be less ambigious. More precisely, I think, is that the analysis is of large telescopes which feed such systems.  Also, I believe to be grammatically correct, it would be: "Tomography" --> "Tomographic"

Abstract, first sent.: "... is challenging to characterize terrestrial planet."  --> " ... though challenging, is a promising path to the characterization of terrestrial planets."  I think is what is meant, not intending to say instead that this approach is somehow "getting in the way."

Same, next sent.: "laser tomography" --> "laser tomographic" and elsewhere, perhaps.

Later: "... unseen spatially frequency ..." --> "... unseen spatial frequencies ..." perhaps.  Also: "LGS" Define this acronymn here, as it is the first time it appears. And: "assit" --> "assist"

As a matter of nomenclature: often/typically the word "asterism" is instead used in the literature for the configuration of laser spots projected on the sky, to be very clear that this is a controllable instrumental parameter, as opposed to a "constellation" which refers to a fixed (and so uncontrollable) assembly of natural guide stars.  If the authors prefer to use the word constellation, then they should clearly define it as understood somehow to be a configurable one, where the term first appears in the text.

Introduction:

First par., first sent.  "... an auspicious ..." is perhaps more clearly "... a promising... " and "project"-->"method", also: "planet" -->"planets" (plural).

At this point, I stopped correcting English; there are various pluralization and minor grammatical mismatches which can be caught by a good editorial proofread.

Later, "LASER" seems to be either an acronym, or was intended to be emphasized.  If the latter, then italics would be better suited.

The usual convention is to capitalize the S in "strehl ratio", as it is a proper name (like, say Hertz or Lyapunov). That's likely to occur many places in the text. An alternative is to stick with SR everywhere, once defined.

This is a good introduction, with seemingly the relevant citations.

Chapter 2:

"Tip/Till" --> "Tip/Tilt"

Just to point out that the assumption of an unobstructed, circular pupil is important no doubt later; no future telescope is like that (nor are perfect coronographs, of course), so the potential impact to calculations must be returned to later.  Also, there doesn't seem to be any justification given for the choice of a single ring of LGS beacons as the asterism studied.  Is it a realistic, or planned configuration for future instruments - or, just computationally easier?  Please explain.

Chapter 3:

"Optimizing" --> "Optimization" I think.

Chapters 4 and 5:

This is easy to follow, and clear.  Figure 3 is particularly illustrative of the advantage to lowered residual RMS wavefront error for multiple LGS asterisms.

Sub-section 5.2.

This is the paper's main result, I believe: "The geometry of the LGS constellation [sic] set the fundamental limitations of the tomography: unseen spatial frequencies."  So this is also my main (only) concern; that the authors explain why they picked this asterism to study, as mentioned in notes on Chapter 2.  If it is somehow mathematically required due to the corresponding choice of an unobstructed circular aperture, then that should be stated; one does not expect so, but it is rather confusing without a reason given.  Here, or later in the conclusions, the authors should also provide some comment on whether the analysis could say something about asterisms other than circular rings, and if those may help improve contrast.  Possibly that's beyond the scope of this analysis, but the reader is no doubt wondering if that is a path worth pursuing, and the authors seem well placed to make some comment or recommendation.

End of Chapter 5, line 224: Here is a minor weakness in the analysis, which is that only 4 nights of profile data are incorporated in setting the parameter limits, using Maunakea profiles.  The reader must wonder if this is representative of that site.  The C_n^2 profiles in Figure 6 do look very diverse: sometimes mostly turbulence at the ground, sometimes mostly at high altitude.  This really matters in the real world, of course.  So, is this representative of other potential 30-m telescope sites?  The authors should briefly qualify the limitations on the results, by providing at least a brief qualitative statement of how representative of any possible site conditions this is.  If they can be quantitative, that would be better, but nothing seems to be said in the current text about this, which leaves the reader pondering how robust the results are.

Chapter 6:

The conclusions are strong and justified, apart from my concern about the choice of asterism - and so making some statement about what broader conclusions can be drawn.  For example, would the authors expect a uniform grid of LGS beacons to provide a better result?  Was that analysis not possible due to the requirement of a unobstructed aperture? Please also state what impact an unobstructed pupil has on the applicability of the results, for the simple reason that no such real telescope is planned. Less serious, the authors should be clear that their conclusions are based on input parameters for a limited dataset on Maunakea.  So, the authors should qualify how well this result may translate to other potential 30-m sites; possibly not well, if for example the profiles of those differ significantly - as one might suspect they could do.  Finally, "0.6 order of magnitude."  Outside astronomy (which of course has to take great care with use of the word magnitude) this reads as a rather odd unit, which I believe means "0.6 X 10 = 6."  If so, I think more typically one would state that as simply a six-fold improvement instead.

Comments on the Quality of English Language

No major issues; please do see my detailed comments on minor issues of tenses, spelling and grammar.  The paper would benefit from careful proofreading/editing.

Author Response

Dear Reviewer,

Thank you very much for taking the time to review our paper and for providing valuable feedback. We have carefully considered your comments and have made revisions to the manuscript accordingly. A point-by-point response to your suggestions has been uploaded as an attachment.

We appreciate your thorough review and constructive criticism, which have undoubtedly helped to improve the quality of our work. Please do not hesitate to contact us if you have any further questions or concerns.

Thank you once again for your time and attention to our manuscript.

Best regards,

Keran Deng

Reviewer 2 Report

Comments and Suggestions for Authors

1. In my opinion, the main point of this manuscript is comparative analysis between LGS AO and LTAO for large ground-based telescopes. Thus, I suggest the authors change the title for satisfying the key point of this manuscript.

2. Some figures are blurred and the authors should redraw them, such as the Figs.1 and 6.

3. Please give the proper units in Figs. 3 and 4 for clear understanding.

Comments on the Quality of English Language

I commend the authors polish their English presentations in detail.

Author Response

(The authors gave the same response as above.)

Reviewer 3 Report

Comments and Suggestions for Authors

Keran Deng et al., “Fundamental Limits on Earth-Like Exoplanet…”

Page 1.  Line 10. Rewrite It is challenging to characterize terrestrial p[lanet using exoplanet imaging with…

This sentence does not belong to Abstract.  

Page 1.  Line 13. Use “derive”  

Page 1.  Line  14. We do not start a sentence with “And”.

“Form “spatially” is not correct. Sentence does not make sense in any case. 

The most accessible rocky planet is nearby low-mass M dwarf stars

Page 1.  Line 15. Sentence starting “By …” is not logical and not clear. Elaborate. 

Page 1, Line 26. Is »rocky planet« the same thing as »habitable planet«? Please explain

This sentence is not correct. Imaging was performed ground arrays in IR light.

Page 1, Line 30. »the angular between” please clarify.

Page 1, Line 31. S is missing in »present”

Page 1, Line 32. »M-type stars present the unavoidable challenges due to stars’ temperature or brightness for direct imaging.”unavoidable challenges – no need to dramatize the obvious

eliminate temperature – it does nort affect imaging.

Page 1, Line 36. 

Add s in »actuator«.

Page 1, Line 38. »The potentially habitable rocky planets around M-type stars will not be imaged with the extreme AO systems because their parent M dwarf stars are not bright enough to ensure that each WFS sub-aperture has a sufficiently high signal-to-noise ratio.«

Provide reference or is this result of the current research?

Comments on the Quality of English Language

Paper requires extensive guessing for comprehension. I recommend rejection to allow authors to improve the quality of presentation (decrease complexity of sentences) and English grammar.

Author Response

(The authors gave the same response as above.)

Round 2

Reviewer 3 Report

Comments and Suggestions for Authors

Conclusions are weak.  What exactly does "Thus, the generalization of our results to other potential 30-meter telescope sites may vary." mean?

Comments on the Quality of English Language

Adequate. Problem is that the absence of facts and results  do not allow clear writing style.